# OpenReview forum: "AutoBaxBench: Bootstrapping Code Security Benchmarking"
_ICLR.cc/2026/Conference — Submitted to ICLR 2026_

### Official Review · Reviewer_utRV · 2025-11-01

**Soundness:** 2
**Presentation:** 3
**Contribution:** 2
**Rating:** 2
**Confidence:** 4

**Summary:**

This paper introduces AutoBaxBuilder, an LLM-based framework for automatically generating code security benchmarks, including new tasks, functional tests, and end-to-end exploits, extending BaxBench. The method employs an agentic orchestration pipeline that iteratively refines scenarios, solutions, and security tests through execution feedback and self-critique. The generated tasks are compared agains BaxBench tests, demonstrating strong agreement and improved exploit coverage. Experiments across state-of-the-art LLMs reveal low secure-pass rates, underscoring the challenge of generating secure code.

**Strengths:**

1. Introduces a fully automated pipeline for constructing code-security benchmarks with minimal human intervention.
2. AutoBaxBench shows strong empirical alignment with BaxBench and deeper exploit coverage.

**Weaknesses:**

1. The paper shows limited novelty beyond automation.
2. The design has potential contamination and bias on benchmark generation and evaluation.

**Questions:**

The paper presents an interesting design; however, in its current form, it is not strong enough to be accepted.

1. The framework mainly extends BaxBench by adding an orchestration layer using stronger LLMs. The algorithmic design follows established LLM-agent paradigms without introducing new security-testing methodologies or vulnerability detection. In essence, the contribution lies in automation rather than conceptual innovation, resulting in limited novelty.

2. AutoBaxBench uses GPT-5 as an orchestration LLM to generate scenarios, test cases and exploits. It iterates on solutions generated by  GPT-5, Claude-4 Sonnet, DeepSeek-R1, and Qwen-Coder 480B. This design introduces the risk of data contamination or model-family bias, since the benchmark generator and evaluated models share overlapping reasoning patterns or prompt structures. A rigorous benchmark should ideally remain model-agnostic, constructed via a fixed, independent toolchain. Could the paper clarify how contamination is mitigated? And what are the performance of GPT-5, Claude-4 Sonnet, DeepSeek-R1, and Qwen-Coder 480B when tested on AutoBaxBench?

3. It remains unclear how many tasks AutoBaxBench implements. The original BaxBench benchmark includes 392 executable tasks across 28 scenarios, while AutoBaxBuilder reports only 40 new “scenarios.” Figure 3 compares AutoBaxBench and BaxBench results side-by-side, but this comparison may be misleading: the total number of test instances are not matched or reported; The difficulty and coverage distributions may not align; And it’s unclear whether the same subset of scenarios is used for comparison.

4. No human study or expert verification is provided for the generated functional and security tests. The paper claims that AutoBaxBench produces “more thorough” or “better” exploits, but this conclusion relies solely on quantitative metrics. A small-scale audit by security experts—evaluating a random sample of generated exploits for correctness, realism, and exploitability—would substantially strengthen the paper’s credibility.

Minor:
– There is an extraneous “s” above “functional requirement” in Figure 1.

---

> ### Author Response · Authors · 2025-11-21
> **Rebuttal**
>
> We thank the reviewer for their insightful remarks and provide answers to their raised questions below.
>
> ### **Why is automating Benchmark generation interesting?**
>
> Benchmark generation is crucial for data-constrained domains such as secure code generation, where available data is scarce and thus requires significant human effort to curate [1,2]. Our method AutoBaxBuilder is one of the first methods to attempt tackling this problem by leveraging LLMs to generate high-quality and rule-based datasets in an almost fully automated pipeline.
>
> [1] He et al, *Instruction Tuning for Secure Code Generation*, ICML ‘24
> [2] Croft et al, *Data Quality for Software Vulnerability Datasets*, ICSE ‘23
>
> ### **Which models of the orchestration, solution, and evaluated LLMs can overlap without causing contamination?**
>
> To reduce the risk of bias and contamination in the evaluation, the evaluation LLMs should not overlap with the LLMs used to generate the benchmark, i.e., the orchestration and solution LLMs. We therefore do not evaluate LLMs that were used in the benchmark creation process on the generated benchmark. We added a note to clarify this in Section 4.
>
> As requested by the reviewer, we report the sec\_pass@1 (pass@1) of the orchestration and solution LLMs used in our main evaluation on AutoBaxBench, the tests generated by our method for BaxBench (AutoBaxBuilder on BaxBench), and BaxBench in the table below. Note that the trend for sec\_pass@1 tends to be strongly decreasing, which intuitively makes sense, since the exploits were generated precisely for the solutions generated by these models.
> In order to reliably evaluate the models GPT-5, Claude 4 Sonnet, Qwen3 Coder 480B and DeepSeek R1, we would therefore recommend to re-run our benchmark generation process with a distinct set of orchestration and solution LLMs, as we did in the newly added Appendix A.2.
>
> |                            | GPT 5       | Claude 4 Sonnet | Qwen3 Coder 480B | DeepSeek R1 |
> |----------------------------|-------------|-----------------|------------------|-------------|
> | AutoBaxBench               | 0.35 (0.62) | 0.34 (0.76)     | 0.12 (0.31)      | 0.18 (0.44) |
> | AutoBaxBuilder on BaxBench | 0.50 (0.74) | 0.38 (0.71)     | 0.25 (0.52)    | 0.35 (0.63) |
> | BaxBench                   | 0.64 (0.71) | 0.53 (0.75)     | 0.34 (0.53)    | 0.39 (0.55) |
>
> Note that for the orchestration and solution LLMs, a potential overlap is unproblematic for contamination. The choice of different models for the solution LLM is only to provide a higher diversity of implementations against which functional tests and security exploits are refined. An extreme case, where tests and exploits were only refined against implementations by the same LLM, would likely result in a worse benchmark, in which test cases are overfit to the specific implementation of the LLM. However, if the model refines its test cases against many implementations, among them its own, we do not observe such issues.
>
>
> ### **Does your method require the use of specific models for operation?**
>
> No, our design is model agnostic. We chose GPT-5 as an orchestration model because it was the most capable model at the time of writing, and several other SOTA coding models of diverse families to increase the diversity of tested solutions. In principle, our method can be done with any set of models, although a strategic choice for the targeted models and diversity is recommended.
>
> As a concrete example, we evaluate using Claude 4.5 Sonnet as orchestration LLM with distinct solution LLMs and provide results of this ablation in Appendix A.2.
>
> ### **Are the results shown in Figure 3 comparable? How many tasks, languages, and frameworks are you evaluating on?**
> Yes, the results shown in Figure 3 are comparable. Overall, we apply the same combination of languages and frameworks to both BaxBench and AutoBaxBench scenarios, resulting in overall $n\times 14$ tasks per model, i.e., 392 tasks on BaxBench and 560 on AutoBaxBench. We also further exactly match the BaxBench evaluation settings, ensuring the comparability to AutoBaxBench results. For further details, please refer to the answer to global question Q2 or Section 4.1 of the revised paper, in which we added a note regarding this topic.
> ### **Please conduct a human verification of the obtained tasks.**
>
> We agree with the reviewer that additional human verification is beneficial. We therefore conducted a systematic, in-depth review. We refer to the review in our extensive report in our answer to global question Q1. We hope that this answer resolves all raised concerns, and kindly request a follow-up question otherwise.
>
> **Further remarks**
>
> We thank the reviewer for their remark about the typo in Figure 1, which has been resolved in the revised paper PDF.

---

### Official Review · Reviewer_y8WS · 2025-11-01

**Soundness:** 3
**Presentation:** 4
**Contribution:** 3
**Rating:** 6
**Confidence:** 4

**Summary:**

The paper presents AUTOBAXBUILDER, an LLM-orchestrated pipeline that automatically creates realistic backend application tasks for secure-coding evaluation, with OpenAPI specs, functional tests, and end-to-end exploits. It validates the generated tests against BAXBENCH, finding close agreement on functionality and stricter security coverage, then releases AUTOBAXBENCH with 40 new scenarios across Easy/Medium/Hard splits. Results show low secure-pass rates for strong models on the hardest split, highlighting a real gap in secure code generation; the whole pipeline is low-cost and scalable.

**Strengths:**

- Realistic, application-level evaluation (multi-endpoint REST backends) instead of toy, single-function settings; language/framework-agnostic via HTTP testing.
- Agentic, execution-in-the-loop generation of scenarios, tests, and exploits with iterative refinement and sanity checks (e.g., OpenAPI validation).
- Tightens security assessment relative to BAXBENCH. Higher exploit coverage and more sensitive tests.
- Scalable & economical: ~2 hours per scenario and a few USD each; three difficulty tiers enable broad model evaluation.
- Clear writing and very strong structure; figures explain the system cleanly.
- Appendix is comprehensive, with detailed statistics, vulnerability analyses, and case studies

**Weaknesses:**

- While the evaluation is extensive, a brief discussion of failure cases where the pipeline struggles (e.g., scenario types it can’t yet reliably generate) would strengthen understanding of current limitations.
- The paper focuses on task generation and evaluation results; a short qualitative example walkthrough (prompt → refinement → exploit) in the main text could make the pipeline’s behavior more concrete for readers.

**Questions:**

- The methodology could be extended beyond web server back-ends, right?
- What about generalizing to other languages like JavaScript (Node.js) or Java? There will be different attack surface being exposed with different languages.

---

> ### Author Response · Authors · 2025-11-21
> **Rebuttal**
>
> We thank the reviewer for their insightful remarks and provide answers to their raised questions below.
>
> ### **Please add more case studies, including showcases where your method fails.**
>
> We refer the reviewer to our answer to global question Q3.
>
>
> ### **Can your methodology be extended to additional settings and languages?**
>
> We want to first highlight the fact that the setup of BaxBench and AutoBaxBench allows writing tests and exploits once, whereas models can be evaluated on practically any language, such as Java and JavaScript. This is important because, as the reviewer correctly points out, different languages expose different attack vectors. We refer to our global answer to Q2 for more details.
>
> Further, it is conceivable to apply our method to using different languages *within* the benchmark generation process, and to apply it to different settings with the same premise, such as binaries with a CLI interface.
>
> ### **Please add a qualitative, running example to ease understanding.**
>
> As suggested by the reviewer, we added a qualitative, running example to Section 3 to help follow the steps of our approach.

---

### Official Review · Reviewer_4Ydm · 2025-11-02

**Soundness:** 2
**Presentation:** 3
**Contribution:** 3
**Rating:** 4
**Confidence:** 3

**Summary:**

This paper extends BAXBench but takes a fully automated approach, where instead of humans designing realistic backend security challenges (scenarios, tests, exploits), the authors propose a pipeline which uses an orchestrator LLM (GPT-5) to invent new backend scenarios, generate functional test cases to ensure the scenario is solvable, and generate security exploits that must break at least one solution (but not all), and to ensure the vulnerability is real, validates via iteration until constraints are satisfied. They claim this allows creation of new benchmark tasks in <2 hours, costing <$10, and that task quality is comparable to human-designed ones.

The system inherits its CWE grounding from BaxBench and claims to cover the same 13 high-severity CWE classes. They even report cases where AutoBAXBench identifies additional CWE types that the original human-authored tests missed (e.g., OS command injection instead of only path traversal).

**Strengths:**

•  important challenge: the high human labour of building security benchmarks like BAXBENCH. Proposing the first end-to-end automated pipeline (AutoBAXBENCH) that generates backend scenarios, functional tests, and security exploits from scratch using agentic LLMs.
•  alignment with existing benchmarks: evaluated on BAXBENCH scenarios, with authors ensuring that the generated benchmarks cover the same 13 high-severity CWE classes. They also show that CWE coverage and per-scenario vulnerability counts match or exceed the original expert-authored tests.
•  quantitative & qualitative evaluation: 6 model families benchmarked across difficulty levels (Easy/Medium/Hard) with pass@1 and sec_pass@1 results reported to be comparable to BAXBENCH. They also include a manual qualitative analysis demonstrating cases where the automated tests detect realistic additional vulnerabilities (e.g., OS command injection in FileSearch) that were not captured by BAXBENCH.
•  produces more thorough attacks than the human baseline on some tasks, eg. detecting memory exhaustion via crafted arithmetic expressions and concurrent loads, whereas BAXBENCH only used basic symbolic misuses.
•  scalability and cost-effectiveness: can produce a new benchmark task (scenario + functional tests + exploit tests) in under 2 hours and <$10 USD

**Weaknesses:**

* limited and unsystematic human validation: The paper  does include a “manual security test analysis” and present a few examples (e.g., OS injection in FileSearch, resource exhaustion in Calculator). This indicates some human inspection occurred. However, this qualitative evaluation is anecdotal (approx. 6 scenarios), lacks methodological transparency (no sampling strategy, number/expertise of reviewers, rubric, or inter-rater agreement), and is not reproducible. In summary, the qualitative evaluation appears to consist only of the authors manually inspecting a small number of generated tests, and as a result, the claim that AutoBAXBench “reproduces or outmatches expert-written tests” is not fully substantiated.
* figures 4 & 5 show correlation between AutoBAXBENCH and BAXBENCH outcomes, but this only demonstrates consistency, not correctness. It's possible that AutoBAXBENCH incorrectly flags additional “vulnerabilities” due to over-strict or misaligned tests.
* risk of falsely accepted pipeline steps: several stages rely on LLM self-judgment and automated execution checks. Without human auditing, it is unclear whether functional tests fully capture the intended behaviour, whether vulnerabilities are genuine, or whether CWE labels are always accurate.

**Questions:**

* who performed the manual analysis? How many experts, what were their credentials, and was it blinded/independent?
* do you have a defined rubric (scenario realism, spec–test alignment, exploit realism, CWE mapping)? Please share it and inter-rater agreement stats if available.
* for the “extra” exploits (the 6 scenarios), were these reproduced against human-written reference implementations (not just LLM solutions)?

The appendix includes a helpful scenario-generation case study (SVG badge) that demonstrates the pipeline’s adaptive test and exploit iteration. However, this single simple example is insufficient evidence by itself.

---

> ### Author Response · Authors · 2025-11-21
> **Rebuttal**
>
> We thank the reviewer for their insightful remarks and provide answers to their raised questions below.
>
> ### **Please conduct a human verification of the obtained tasks.**
>
> We agree with the reviewer that additional human verification is beneficial. We therefore conducted a systematic, in-depth review. We refer to the review in our extensive report in the answer to global question Q1. We hope that this answer resolves all raised concerns, and kindly request a follow-up question otherwise.
>
> ### **Please add more case studies, including showcases where your method fails.**
>
> We refer the reviewer to our answer to global question Q3.

---

### Official Review · Reviewer_RoZm · 2025-11-03

**Soundness:** 2
**Presentation:** 2
**Contribution:** 2
**Rating:** 2
**Confidence:** 5

**Summary:**

As LLMs become widely used in software engineering, assessing the correctness and security of their generated code is increasingly important, yet existing manually crafted benchmarks are limited and unsustainable. To address this, the paper introduces AutoBaxBench, a fully automated framework that generates security benchmarking tasks and tests using LLMs, incorporating plausibility checks and end-to-end exploit generation. Evaluations show that AutoBaxBench produces high-quality benchmarks comparable to expert-created ones, enabling new task generation in under two hours at a cost of less than $10.

**Strengths:**

1. Addresses an important and timely problem of generating security benchmarking tasks and tests using LLMs. These in turn can be used to automatically evaluate correctness and security of LLM-generated code against those tasks and tests.

2. The evaluation shows that AutoBaxBench is on par with the manually created predecessor, BaxBench, in terms of correctness and security trends for a variety of different LLMs.

**Weaknesses:**

1. The presentation of the paper needs significant improvement. I found the main Algorithm 1 to be highly incomprehensible
- What is the desired goal of Steps 2 and 3?  How do LLMs help accomplish those goals? Even if LLMs can only approximately achieve those goals, it would be good to know what the ideal outcome is.
- Why are two different LLMs used for M and M_s? Is the intention to use the best model (like GPT-5) for M, but weaker LLMs for M_s?
- Is the pseudo code of various functions in Algorithm 1 (such as refine_tests, refine_solutions, etc.) provided somewhere? I was only able to find informal descriptions in Section 3.
- A running example showing what is generated at each step would also be helpful.

2. AutoBaxBench would be most useful when one wants to improve a specific LLM.  How can one achieve this? Suppose the target LLM is M_c.  How should one pick M and M_s in Algorithm 1, and why?

3. There is hardly any discussion in the paper about the kinds of languages, tasks, and correctness/security properties that AutoBaxBuilder supports. Section 2 notes "Each such combination defines a language-independent task, which can readily be evaluated in 14 frameworks across 6 programming languages." What are these frameworks and languages?

Minor: the terminology AutoBaxBuilder vs. AutoBaxBench is confusing. I think you are proposing the former, a benchmark builder, and you have generated one instance of the benchmark called AutoBaxBench. But the paper uses these terms interchangeably, e.g. calling the framework itself AutoBaxBench in the abstract.

**Questions:**

Please see weaknesses.

---

> ### Author Response · Authors · 2025-11-21
> **Rebuttal**
>
> We thank the reviewer for their insightful remarks and provide answers to their raised questions below.
> ### **What is the desired goal of Steps 2 and 3? How do LLMs help accomplish those goals?**
> The desired goals of Steps 2 and 3 are to generate strongly differentiating functional tests and security exploits, e.g., test cases that allow distinguishing faulty implementations or insecure ones from correct and secure ones, respectively.
>
> Concretely, the output of Step 2 should be functional tests that enable differentiating between backend implementations that follow the given endpoint specification and correctly implement the described logic, and implementations that do not. In the Calculator scenario, such a functional test should verify that the API endpoint `/calculator` is present and correctly evaluates a range of provided arithmetic expressions.
>
> The output of Step 3 should be end-to-end exploits that should be able to expose present vulnerabilities in a wide variety of implementations that exhibit the vulnerability. For example in the Calculator example, such an exploit can involve sending `system(“rm -rf /”)` to the evaluation endpoint. In Python implementations that simply defer to the built-in `eval` method, this would lead to a successful OS Injection. However, the exploit is not ideal, since it will not work on implementations written in other languages.
>
> We adjust Section 3 to highlight these expected outcomes and add a running example. As the generated code has a considerable length, we defer a concrete case study with the generated code to Appendix B.1.
> ### **Why are different LLMs used as orchestration and solution LLMs?**
> The goal of this separation is to allow using several LLMs as solution LLMs. This is desirable to increase the diversity of implementations on which Step 2 and Step 3 will be evaluated, which again aim to handle a diverse range of implementations during evaluation. Having a wide range of implementation candidates to test during the construction helps adapt tests to handle this diversity. We add a note highlighting this to Section 3.
>
> ### **Please provide pseudocode for your subroutines**
> We thank the reviewer for remarking that this was missing and added the pseudocode for the requested functions, together with the concrete used prompts in Appendix C.
> ### **How can AutoBaxBench be used to create a benchmark for a specific target model?**
> In order to improve a specific model, we argue that it should be uninvolved in the benchmark creation process in order to avoid biases or contamination. As such, one should choose unrelated models (ideally from a different model family) for the orchestration and solution LLMs in Algorithm 1.
>
> As a concrete example of why this is important, we add an ablation in which Claude Sonnet 4.5 generates scenarios and test cases. We then evaluate Claude Sonnet 4.5 on both datasets. We observe that the performance of Claude Sonnet 4.5 on its own tests is suspiciously high (95% pass@1) whereas performance on the GPT-5 generated benchmark is markedly lower at 81%. For more details on this ablation, we refer to Section 4.4 and Appendix A.2 in the revised paper.
> ### **How many languages and frameworks are you evaluating on?**
> Please refer to the global answer to Q2.
>
> **Further remarks**
>
> We thank the reviewer for highlighting the differences in our usage of AutoBaxBuilder vs AutoBaxBench. As suggested, we made the usage of AutoBaxBuilder vs AutoBaxBench consistent throughout the revised paper.

---

### Author Response · Authors · 2025-11-21
**Global Response to Reviewers**

We would like to thank all reviewers for the valuable feedback and insightful questions. We are happy to see that many reviewers agree that our work tackles an important and timely research question, and acknowledge that our results quantitatively align with previous, manually created benchmarks.

We observe that many reviewers requested a manual review of the results. We provide an extensive review and provide its details in the answer to global question Q1. Moreover, we clarify the exact number of tasks for evaluation in Q2 and provide more case studies in the appendix of the revised paper. Finally, we answer all individual questions in the comments to the respective reviews.

---

> ### Author Response · Authors · 2025-11-21
> **Expert Verification of Results (Q1)**
>
> ### (4Ydm, utRV) **Q1: Please provide a manual verification of the benchmark by human experts**
>
> We conduct an expert evaluation of the generated scenarios to assess the quality of the generated scenarios, test cases, and security exploits. For this task, we recruited four security experts, including Master's, PhD-level education, and industry experience at a cybersecurity company.
> We sample 6 scenarios from BaxBench [1] and 6 Scenarios from our generated AutoBaxBench, where for the latter we sample 2 scenarios from the Easy, Medium, and Hard subsets each.
>
> For every scenario, the experts are provided with the generated scenario specification, functional tests, and security tests. They are instructed to independently assess the validity and quality of the scenario, each functional test, and each security test.
>
> We report the questionnaire and a summary of the results of our human evaluation in Appendix A.4 of the revised paper in Table 4, including average scores and inter-rater agreement. The results indicate that the majority of the generated scenarios and tests were rated positively by the experts, underlining the effectiveness of AutoBaxBuilder in producing high-quality benchmark scenarios.
>
> **High realism and low ambiguity**
> Scenario-related metrics highlight some concerns about ambiguity in the scenario specifications, rating $79\%$ of scenarios as unambiguous. This is often due to missing edge cases in the OpenAPI specification. However, the scenarios were still generally rated as realistic at $83\%$. This area generally shows low inter-rater agreement, as the rating is often subjective.
>
> **Overall correct functional tests**
> Notably, the functional tests received particularly high scores with strong agreement, with over $98\%$ of the tests being rated as correctly implementing the intended functionality and $99.4\%$ matching the OpenAPI specification. This suggests that AutoBaxBuilder is highly effective at extracting and testing functional requirements from the specification.
>
>
> **Concerns about coverage and CWE classification**
> The security tests received high scores around $97\%$ for sensibility and overall soundness, indicating that the pipeline correctly discards and modifies exploits that are fundamentally flawed. The high inter-rater agreement adds confidence that the false positives are limited. Meanwhile, only $81\%$ of exploits are marked to report a correct CWE classification. This is due to fundamental ambiguity in how to classify CWEs, which is also visible in the low inter-rater agreement. Moreover, the exploit coverage is overall rated low at only $71\%$ of exploits being marked for sufficient coverage. The experts thus frequently express concern that the exploits would not generalize well to new implementations. While the score is moderate, the low inter-rater agreement indicates that even human experts disagree on the exact precision of some exploits.
>
> The soundness score of $96.77\%$ implies an upper bound on the sec\_pass@1. It could increase by at most a margin of $3.23\%$ if all false positives were eliminated, providing an upper bound on improvement achievable through fixing unsound exploits.
>
> Qualitatively, the experts remarked on the ambiguity of CWE-400-related exploits. When inspecting the union of all expert reports, all CWE-400 related reports are marked as unsound by at least one expert. Further, for some instances, AutoBaxBuilder generates less diverse attack vectors than desired, resulting in mixed results compared to the prior study on BaxBench.
>
> [1] Vero et al, *BaxBench: Can LLMs Generate Correct and Secure Backends?*, ICML ‘25

---

> ### Author Response · Authors · 2025-11-21
> **Clarification on Tasks and Case Studies (Q2, Q3)**
>
> ### (RoZm, UtRV, y8WS) **Q2: Please clarify how many tasks, languages, and frameworks you evaluate on.**
> The generated tasks define a general REST backend that exposes a unified API. The test cases and exploits target these API endpoints. As such, the generated scenarios of AutoBaxBench can be used to evaluate capabilities of models in any language that supports building REST backends.
> In our evaluations, both in Section 4.2 and 4.3, we mirror the evaluation setting of BaxBench [1], and test model-generated code in 14 frameworks across 6 languages, resulting in a total of 392 and 560 evaluation tasks for BaxBench and AutoBaxBench, respectively. We add a note to Section 4.1 to clarify this point.
>
> [1] Vero et al, *BaxBench: Can LLMs Generate Correct and Secure Backends?*, ICML ‘25
>
> ###  (4Ydm, y8WS) **Q3: Please add case studies that showcase failure modes of your method.**
>
> In addition to the end-to-end generation example in Appendix B.1, we add three concrete examples of incorrect generated functionality tests and security exploits to Appendix B.2. The examples showcase three different failure modes: Generating exploits that miss vulnerabilities, because they overfit to a reference solution, exploits that report non-existent vulnerabilities because they misinterpret the OpenAPI specification, and similarly incorrect functional tests due to OpenAPI misinterpretation.

---

> ### Author Response · Authors · 2025-11-21
> **List of Changes to the Revised Paper**
>
> We provide a list of all changes to the paper below. The changes are highlighted in the revised paper in $\textcolor{teal}{\text{teal}}$ for ease of recognition. Where entire sections or figures were added, we highlight the titles and captions.
> - Section 3:
>   - We clarify the pseudocode for AutoBaxBuilder
>   - We add notes to highlight our design goals.
>   - We add a running example
> - Section 4:
>   - We clarify the total number of scenarios
>   - We exclude exploits raising CWE-400 from our main quantitative results
>   - We add Claude 4.5 Sonnet to the evaluation
>   - We add the results of our human validation of the generated code
>   - We add two ablations on the choice of model for benchmark generation. (Sec 4.4)
> - Section 5: We rewrite the related work section for conciseness.
> - Appendix A: Now comprises the experimental details and manual verifications by one paper author. Further
>   - App. A.5 We add the details on our expert evaluation
>   - App. A.6 We compare the change in results when including CWE-400
> - Appendix B:
>   - App. B.1 We provide more details on the end-to-end generation example, matching the example in Section 3.
>   - App. B.2. Showcases three distinct failure modes of our method
> - Appendix C: We added pseudocode of subroutines and connected them to the prompts

---

### Author Response · Authors · 2025-12-02
**Summary of the discussion phase**

We thank the reviewers and AC to take the time and review our paper. We would like to provide a summary of our discussion until the closing of the discussion phase at the end of last week.

The main request from the reviewers was a manual verification of our generated data. We conducted an extensive manual review using 4 security experts, showing that our LLM-generated benchmark indeed matches prior similar benchmarks with respect to coverage and reliability of generated tests. Meanwhile, we cautiously discard potentially flaky tests and re-evaluate the method, showing that the previously observed trends still hold. We provide a summary of our findings in our global response, and a detailed description in the uploaded PDF revision, highlighted in $\textcolor{teal}{\text{teal}}$.

Additional questions have been addressed in the individual reviews. There has been no answer or score change since our posting of the findings.

---

### Meta-Review · Area_Chair_MsrN · 2025-12-20

**Summary:**

The authors proposed a new benchmark to evaluate security in code generation tasks. Specifically, the authors introduced AutoBaxBench, a new framework to generate tasks and tests from scratch, including a pipeline with fine-grained plausibility checks by using LLMs to construct functionality tests and security-probing exploits. The authors provided qualitative and quantitative evaluation against tasks constructed by human experts.

**Reviewer Concerns:**

1. There is writing issue in presenting the method, leading to some confusion in technical details e.g. the purpose of step 2 and 3 in the data pipeline or the notation in Algorithm 1.
- The authors has tried to addressed this issue during the rebuttal, including added details in the manuscript. Overall, I found their response addressed this reviewer concern.

2. There is limited manual human validation. Some manual inspection is done in the paper but they are mainly unsystematic and not reproducible according to reviewers.
- The authors provided a global response with verification conducted human experts. However, the sampled data is rather small with only 6 scenarios from AutoBaxBench.

3. There is some novelty concerns in the framework a it is mostly automation by LLMs without introducing new methodology to improve security testing or vulnerability detection. Since the method are heavily depending on LLMs during many stages in the pipeline, there is a risk of data biases and contamination in the generated data.

**Reviewer Scores:**

Since the authors have engaged quite significantly during the rebuttal, there might be some change in the scores:
- Reviewer RoZm: would improve the score from 2 to 3 or 4 based on the authors' response on presentation issues and the details of the method
- Reviewer 4Ydm: might not improve the score (4) because the main concern on manual validation still persists
- Reviewer y8WS: would keep the same positive score (6)
- Reviewer utRV: might improve the score but at most to 3 because of the main concerns on novelty and human validation

---

### Decision · Program_Chairs · 2026-01-26

Reject